# Clinical Analysis of Early-Stage Pancreatic Cancer and Proposal for a New Diagnostic Algorithm: A Multicenter Observational Study

**DOI:** 10.3390/diagnostics11020287

**Published:** 2021-02-12

**Authors:** Juri Ikemoto, Masahiro Serikawa, Keiji Hanada, Noriaki Eguchi, Tamito Sasaki, Yoshifumi Fujimoto, Shinichiro Sugiyama, Atsushi Yamaguchi, Bunjiro Noma, Michihiro Kamigaki, Tomoyuki Minami, Akihito Okazaki, Masanobu Yukutake, Yasutaka Ishii, Teruo Mouri, Akinori Shimizu, Tomofumi Tsuboi, Koji Arihiro, Kazuaki Chayama

**Affiliations:** 1Department of Gastroenterology and Metabolism, Graduate School of Biomedical and Health Sciences, Hiroshima University, Hiroshima 734-8551, Japan; serikawa@hiroshima-u.ac.jp (M.S.); yishii@hiroshima-u.ac.jp (Y.I.); tsuboitomo@hiroshima-u.ac.jp (T.T.); chayama@mba.ocn.ne.jp (K.C.); 2Department of Gastroenterology, Onomichi General Hospital, Onomichi 722-8508, Japan; kh-ajpbd@nifty.com (K.H.); a.shimizu313@gmail.com (A.S.); 3Department of Gastroenterology, Hiroshima Memorial Hospital, Hiroshima 730-0802, Japan; eguchi@kkrhiroshimakinen-hp.org; 4Department of Gastroenterology, Hiroshima Prefectural Hospital, Hiroshima 734-8530, Japan; tamito0611@gmail.com; 5Department of Gastroenterology, Hiroshima General Hospital, Hatsukaichi 738-8503, Japan; fujiyt@yc4.so-net.ne.jp; 6Department of Gastroenterology, Saiseikai Hiroshima Hospital, Aki 731-4311, Japan; sugishin111@gmail.com; 7Department of Gastroenterology, National Hospital Organization Kure Medical Center and Chugoku Cancer Center, Kure 737-0023, Japan; yamaguchi.atsushi.uc@mail.hosp.go.jp; 8Department of Gastroenterology, Kure Kyosai Hospital, Kure 737-8508, Japan; b-noma@kure-kyosai.jp; 9Department of Gastroenterology, Saiseikai Kure Hospital, Kure 737-0921, Japan; m_kamigaki05@yahoo.co.jp; 10Department of Gastroenterology, National Hospital Organization Higashihiroshima Medical Center Affiliation, Higashihiroshima 739-0041, Japan; t.minami@beach.ocn.ne.jp; 11Department of Gastroenterology, Hiroshima Red Cross Hospital & Atomic-bomb Survivors Hospital, Hiroshima 730-8619, Japan; ak.from.coast@gmail.com; 12Department of Gastroenterology, Hiroshima City Asa Citizens Hospital, Hiroshima 731-0293, Japan; m_yukutak@yahoo.co.jp; 13Department of Gastroenterology, Chugoku Rosai Hospital, Kure 737-0193, Japan; teruo0914@chugokuh.johas.go.jp; 14Department of Anatomical Pathology, Hiroshima University Hospital, Hiroshima 734-8551, Japan; arihiro@hiroshima-u.ac.jp

**Keywords:** early diagnosis, pancreatic ductal adenocarcinoma, multicenter study, endoscopic retrograde cholangiopancreatography, endoscopic ultrasonography

## Abstract

Early diagnosis of pancreatic ductal adenocarcinoma (PDAC) is challenging but essential for improving its poor prognosis. We established a multicenter study to clarify the clinicopathological features, and to propose new algorithm for early diagnosis of PDAC. Ninety-six patients with stage 0 and IA PDAC were enrolled from 13 high-volume centers. Overall, 70% of the patients were asymptomatic. The serum pancreatic enzyme levels were abnormal in half of the patients. The sensitivity of endoscopic ultrasonography (EUS) for detecting small PDAC was superior to computed tomography and magnetic resonance imaging (MRI) (82%, 58%, and 38%, respectively). Indirect imaging findings were useful to detect early-stage PDAC; especially, main pancreatic duct stenosis on MRI had the highest positive rate of 86% in stage 0 patients. For preoperative pathological diagnosis, the sensitivity of endoscopic retrograde cholangiopancreatography (ERCP)-associated pancreatic juice cytology was 84%. Among the stage IA patients, EUS-guided fine-needle aspiration revealed adenocarcinoma in 93% patients. For early diagnosis of PDAC, it is essential to identify asymptomatic patients and ensure close examinations of indirect imaging findings and standardization of preoperative pathological diagnosis. Therefore, a new diagnostic algorithm based on tumor size and imaging findings should be developed.

## 1. Introduction

Patients with pancreatic ductal adenocarcinoma (PDAC) are generally diagnosed at the advanced stage and generally have a very poor prognosis. The American Cancer Society estimated that in 2019, approximately 56,770 patients were diagnosed with PDAC in the USA and 45,750 would be dead due to the disease [1]. The 5-year survival rate for patients with PDAC is only 6 to 8% in the USA [2]. The national Cancer Center Japan reported that 35,390 individuals died from PDAC in Japan in 2018 [3]. In contrast, according to an analysis of the recent Japan Pancreatic Cancer Registry, which included more than 350 high-volume centers, the 5-year survival rate of PDAC patients with a tumor diameter <10 mm was 80.4%, and that for stage 0 PDAC patients defined by the Union for International Cancer Control (UICC) was 85.8% [4]. However, the cancer registry has reported that UICC stage 0 and IA patients accounted for only 1.7% and 4.1%, respectively, of all PDAC patients [4]. These findings indicate that although early diagnosis is essential for improving the prognosis of patients with PDAC [5], it remains to be a great challenge.

Recently, several reports on imaging features of early-stage PDAC have been published [6,7,8,9]. Indirect imaging findings, such as dilatation or stenosis of main pancreatic duct (MPD), pancreatic cysts, and local fatty changes were crucial signs of early-stage PDAC [6,7,8,9]. However, no consensus on the diagnostic algorithm and pathological diagnosis has been established yet. Additionally, there are typically only few patients with early-stage PDAC in each single institution, and it has proved difficult to study these.

These previous observations prompted us to establish this multicenter study to clarify the clinicopathological features of early-stage PDAC and to propose an effective diagnostic algorithm for detecting early-stage PDAC based on the data from this study.

## 2. Materials and Methods

### 2.1. Patients

This study was a retrospective, multicenter, observational study. From January 2000 to September 2020, patients with early-stage PDAC from Hiroshima University Hospital and 12 affiliated high-volume centers were enrolled. Early-stage PDAC was defined as patients with stage 0 (high-grade pancreatic intraepithelial neoplasia/pancreatic carcinoma in situ (PCIS)) and stage IA (invasive carcinoma with tumor diameter of <20 mm localized within the pancreas, along with the absence of regional lymph nodes metastasis and distant metastasis) based on post-operative pathological classification according to the seventh edition of the Japanese Classification of Pancreatic Carcinoma [10]. Patients with intraductal papillary mucinous neoplasm (IPMN) concomitant PDAC were included, but patients with high-grade IPMN or IPMN-derived invasive cancer showing a histologic transition between IPMN and PDAC were excluded. In stage 0 cases, an experienced pancreatic pathologist not affiliated with the participating institutions reevaluated each pathologic specimen for tumor characteristics, including validity of the diagnosis, histological type, invasion of the tumor, and exclusion of IPMN and IPMN-derived invasive cancer, regardless of previous diagnosis that had been made at each of the institutions.

We retrospectively reviewed data on the following items: (1) patient characteristics, (2) reasons for medical examination, (3) blood tests and imaging findings, and (4) preoperative pathological examination. 

### 2.2. Imaging and Pathological Diagnosis

#### 2.2.1. Imaging Diagnosis

Depending on the facilities available at each institution, CT (computed tomography) was performed using a multidetector row (from 64 to 320 slices). CT included unenhanced and contrast material-enhanced biphasic imaging, the latter of which comprised arterial and portal phases. Magnetic resonance imaging (MRI) was performed using 1.5 or 3 T. For endoscopic ultrasonography (EUS), we used a radial echoendoscope (GF-UE260 and GF-UM2000; Olympus Medical Systems, Tokyo, Japan), which was superior for observing the MPD in the long axis, equipped with processors (EU-ME1and EU-ME2; Olympus Medical Systems, α-10 and F75; Hitachi-Aloka, Tokyo, Japan). If the pancreatic tail was poorly observed, linear echoendoscope (UCT-260; Olympus Medical Systems) was also used as needed. According to the previous reports [6,7,8,9], indirect imaging findings were defined as MPD stenosis or dilatation, pancreatic cysts, localized pancreatic atrophy, and hypoechoic area surrounding the MPD stenosis.

#### 2.2.2. Pathological Diagnosis

For preoperative pathological examinations, we performed endoscopic retrograde cholangiopancreatography (ERCP) using a video duodenoscope (JF-260V and TJF-260V; Olympus Medical Systems). After pancreatography, we inserted a 0.025 inch guidewire (VisiGlide2; Olympus Medical Systems, Wrangler; PIOLAX Medical Device, Yokohama, Japan, Jagwire^TM^; Boston Scientific, Marlborough, MA, USA) into the MPD. We chose patients with localized stenosis or distal dilatation of the MPD among those undergoing ERCP, and an endoscopic nasopancreatic drainage (ENPD) catheter was placed into the MPD for serial pancreatic juice aspiration cytologic examination (SPACE) [11]. We used a 4 or 5-Fr ENPD catheter (Gadelius Medical, Tokyo, Japan) to collect pancreatic juice for up to six times on one day and subsequently removed the ENPD catheter. We also performed brushing cytology in some patients whose pancreatography showed localized stenosis, and the brushing cytology catheter (RX Cytology Brush; Boston Scientific) was reliably placed in the MPD, caudal to the stenosis. The brush was inserted into the MPD of interest over the guidewire and positioned distal to the stricture. It was advanced from the sheath to a point proximal to the stricture and moved across the stricture in a to-and-fro manner 10 to 15 times. After that, the pancreatic juice in the catheter was flushed with saline for collection.

When obvious lesions were observed, or when ERCP-associated pancreatic juice cytology (PJC) results were negative, EUS-guided fine-needle aspiration (EUS-FNA) was performed using a linear echoendoscope (UCT-260; Olympus Medical Systems) with a 22 or 25 gage needle (Expect^TM^ and Acquire^TM^; Boston Scientific, EZ Shot 3 Plus; Olympus Medical System, EchoTip^TM^; COOK Medical, Bloomington, IN, USA). These examinations were performed under the supervision of specialists who had experienced more than 100 EUS-FNA and 500 ERCP procedures for a period of more than ten years, according to the standard imaging procedures of Hiroshima University Hospital.

### 2.3. Statistical Analysis

We performed statistical analysis using the JMP v14.0 software (SAS Institute, Chicago, IL, USA). The Kaplan–Meier method was used to estimate cumulative survival. With respect to two-tailed tests, Pearson’s χ^2^ test was used to identify statistically significant differences. *p* values < 0.05 were considered statistically significance.

## 3. Results

### 3.1. Patient Characteristics

Table 1 shows the clinical characteristics of all patients. Forty patients with stage 0 PDAC and 56 patients with stage IA PDAC were enrolled. There were 47 men and 49 women, with the mean age of 71 years (range: 39–88). The location of PDAC was pancreatic head in 30 patients, pancreatic body in 50 patients, and pancreatic tail in 12 patients. Multiple lesions were observed in four patients. Each risk factor of PDAC was 28% for pancreatic cysts including IPMN, 27% for diabetes mellitus (DM), 26% for alcohol consumption (intake of more than 37.5 g of ethanol/day), 31% for tobacco use, 3.1% for obesity (body mass index >30 kg/m^2^), 6.5% for a history of acute pancreatitis, 7.4% for chronic pancreatitis, and 2.4% for a family history of pancreatic cancer (at least one PDAC patient among the first-degree relatives). Seventy-one percent of the patients had one or more of the above risk factors, with a mean of 1.3.

### 3.2. Reasons for Medical Examination

Twenty-seven patients (28%) came to the first medical examination with some symptoms, whereas 67 patients (70%) were asymptomatic. The asymptomatic patients comprised those with abnormalities detected during medical health check-ups (39%), those with abnormalities incidentally detected during screening or surveillance for other diseases (52%), and those with changes found during the follow-up of pancreatic diseases (9%).

Of the 26 patients with abnormalities detected during the medical health check-ups, 18 (69%) were detected by US (ultrasonography), two (7.7%) were detected by CT, and three patients (11%) exhibited elevated serum pancreatic enzyme levels. The abnormal findings detected by US included tumors in four patients and indirect imaging findings in 14 patients. The abnormal findings detected by CT included a pancreatic tumor in one patient and indirect imaging finding in one patient.

Among the 35 patients in whom any abnormalities were incidentally detected during examination or follow-up for other diseases, two (5.7%) were detected by US, 19 (54%) were detected by CT, two (5.7%) exhibited elevated serum pancreatic enzyme levels, and one patient (2.9%) exhibited elevated serum tumor marker level. All of the abnormal findings detected by US were indirect imaging findings. The abnormal findings detected by CT included tumors in four patients and indirect imaging findings in 15 patients. Of the other four patients, three patients had pancreatic abnormalities noted on imaging examination other than US or CT, and one patient was diagnosed with PCIS in the post-operative specimen performed for extrahepatic cholangiocarcinoma.

Six patients were detected pancreatic abnormalities during follow-up for pancreatic disease, including pancreatic cysts, and acute and chronic pancreatitis. All patients had been followed up every six months with either CT, EUS, or MRI and measurement of serum tumor markers including carcinoembryonic antigen (CEA) and carbohydrate antigen 19-9 (CA19-9) (Table 2).

### 3.3. Examination

#### 3.3.1. Blood Tests

The serum pancreatic enzyme levels were abnormal in 49% patients. Increased lipase levels were observed in 48% patients, increased pancreatic amylase levels secreted from pancreas in 30% patients, and increased elastase 1 levels were observed in 29% patients. As for serum tumor markers, the levels of carcinoembryonic antigen, carbohydrate antigen 19-9, duke pancreatic monoclonal antigen type 2, and s-pancreas-1 antigen were elevated in 6.9%, 27%, 17%, and 19% patients, respectively. The positive rate of all tumor markers in patients with stage IA was higher than that of patients with stage 0 (Table 3).

#### 3.3.2. Imaging Diagnosis

Imaging diagnostic data are shown in Table 4. For the stage IA patients, tumors were detected in 53% (20/38) patients by US, 58% (31/53) patients by CT, 38% (15/39) patients by MRI, and 82% (45/55) patients by EUS. Of the 22 patients without tumorous lesions detected by CT and of the 19 patients without tumorous lesions detected by MRI, the use of EUS enabled the detection of a pancreatic mass in 64% and 58%, respectively. EUS enabled a significantly higher rate of tumor detection than CT or MRI (*p* < 0.01).

For the stage 0 patients, the vast majority of abnormal findings reflected MPD abnormalities, such as stenosis and dilatation. US, CT, and MRI including magnetic resonance cholangiography (MRCP), and EUS showed MPD dilatation in 62% (8/13), 72% (28/39), 78% (28/36), and 78% (31/40) patients, respectively. MRI including MRCP and EUS showed MPD stenosis in 86% (31/36) patients and 73% (29/40) patients, respectively. The detection rate of MPD abnormalities by MRCP was similar to that by ERCP. In addition, localized pancreatic tissue atrophy was observed in 31% (12/39) patients on CT and 10% (4/40) patients on EUS.

### 3.4. Preoperative Pathological Diagnosis

Preoperative pathological diagnostic data are shown in Table 5. For 38 patients with stage 0 and 48 patients with stage IA, ERCP-associated PJC was performed. The confirmation of malignancy with ERCP-associated PJC was 87% and 81% for patients with stage 0 and stage IA, respectively. During the ERCP session, single cytology showed a sensitivity of 44% (15/34) for patients with stage 0 and 32% (11/34) for patients with stage IA, while brushing cytology of the MPD stenosis showed a sensitivity of 100% (3/3) for stage 0 and 65% (11/17) for stage IA. SPACE using ENPD catheter showed a sensitivity of 83% (29/35) for patients with stage 0 and 73% (29/40) for patients with stage IA. Post-ERCP pancreatitis (PEP) accounted for 7% of all patients, all of which were mild according to Cotton’s criteria [12] and resolved with conservative treatment. Among the stage IA patients, 15 patients (27%) underwent EUS-FNA, which revealed adenocarcinoma in 93% (14/15) patients. The preoperative diagnosis of PDAC was obtained pathologically in 79 patients (82%).

### 3.5. Prognosis

The 5-year overall survival (OS) rate was 87% for stage 0 and 71% for stage IA, while the 10-year OS rate was 57% for stage 0 and 41% for stage IA. The 5-year disease-specific survival (DSS) rate was 94% and 82% for stage 0 and stage IA, respectively, and the 10-year DSS rate was 81% and 51% for stage 0 and stage IA, respectively. The 5-year recurrence-free survival (RFS) rate was 92% and 68% for stage 0 and stage IA, respectively, and the 10-year RFS rate was 83% and 49% for stage 0 and stage IA, respectively. The DDS and RFS rates of stage 0 patients were significantly superior to stage IA patients (*p* = 0.021 and 0.013, respectively) (Figure 1). Twenty patients experienced a recurrence of PDAC: three patients were stage 0 and 17 patients were stage IA at the initial surgery. Recurrence was detected in the remnant pancreas in 13 patients (three patients in stage 0, 10 patients in stage IA), in the lung in two patients, in the liver in three patients, and local recurrence was observed in one patient.

## 4. Discussion

In this study, we investigated the process from identifying patients with early-stage PDAC to scrutinizing the pancreas by imaging examination and pathological diagnosis.

First, this study indicates that efficient identification of asymptomatic PDAC patients is important for early diagnosis. The Clinical Practice Guidelines for Pancreatic Cancer (CGL) issued by Japan Pancreas Society (JPS) in 2019 have identified risk factors for PDAC listed in Table 1 [13]. In this study, 71% patients exhibited at least one risk factor and about 30% patients had DM, tobacco use, and pancreatic cysts. Singhi et al. reported that the onset or exacerbation of DM triggered the diagnosis of early-stage PDAC [14]. In this study, eight patients were also found to be triggered by the onset or exacerbation of DM. Exacerbation of DM was defined as an increase in HbA1c of 1% or more within six months. Development of the condition in patients with DM should be carefully monitored.

In asymptomatic patients, 38% were identified at medical health check-ups, and abnormalities noted on US were the most common. Of these, 78% of patients were detected with indirect imaging findings. Hanada et al. established a social diagnostic project on early-stage PDAC [15,16]. Doctors specialized in PDAC have recommended general practitioners to perform US for patients exhibiting the risk factors of PDAC. Finally, the project achieved to improve surgical resection rate, increase in early-stage PDAC patients, and improve 5-year survival rates of up to 20% [16]. Collectively, we suggest US screening of patients exhibiting the risk factors and recommend further examination to not only pancreatic tumors but also indirect imaging findings.

Regarding blood tests, the levels of tumor markers may generally be low in the early-stage PDAC in this study [17]. However, serum CA19-9 was elevated in 38% patients in stage IA in this study. It has been reported that there was no correlation between the tumor diameter and the serum CA19-9 level except for PDAC with a diameter of more than 6 cm [18], and it is considered necessary to measure serum CA19-9 level even for a small pancreatic tumor. Furthermore, serum pancreatic enzyme levels were abnormal in nearly half of patients. Therefore, we recommend that abnormalities of serum pancreatic enzyme levels should be evaluated to increase the likelihood of early PDAC diagnosis.

Before forming a tumor, small PDACs may increase the intraductal pressure of the pancreatic ducts and could lead to the dilatation of the caudal or branched pancreatic ducts, resulting in the formation of cystic lesions [19,20]. Therefore, it is important to establish the diagnosis also in patients where no direct detection of tumors can be found. In previous reports, the sensitivity of the MPD stenosis in early-stage PDAC detected by US and CT was 20% and 50%, respectively [21,22]. In this study, MPD stenosis on MRI including MRCP had the highest positivity rate (81%), especially in stage 0 patients (86%). This observation suggested that MRCP should be useful for detecting the MPD stenosis. Although the CGL issued by JPS in 2019 treats CT, MRI, and EUS equally as the second imaging modalities for close examination [13], we would like to recommend MRI including MRCP as a second imaging modality based on this study.

Furthermore, we would like to recommend EUS as the third imaging modality. In this study, EUS enabled a significantly higher rate of tumor detection in stage IA patients than CT or MRI, and more than half of the patients in which a mass was not detected by CT or MRI had a tumor detected by EUS. It has been reported that EUS has a diagnostic sensitivity of 94.4% for detecting small PDAC (<20 mm) [23]. Yasuda et al. reported that of 132 patients with risk factors for PDAC without mass detected on CT, pancreatic tumors in three patients were detected by EUS [24]. The CGL issued by JPS in 2019 stated that EUS should be performed in institutions having highly skilled operators. [13]. In this study, the EUS skills were harmonized by the training program at Hiroshima University Hospital. Collectively, we strongly recommend using EUS for this purpose in institution where high-level skills are available.

In this study, patients with PDAC were frequently diagnosed during the screening or surveillance for other diseases by CT and indirect imaging findings were observed in 43%. In recent years, it has been reported that patients with chronic liver disease who underwent surveillance for hepatocellular carcinoma may be diagnosed with PDAC at an early stage [25]. In addition to MPD abnormalities, localized pancreatic atrophy observed on CT may be one of the important indirect findings of PCIS [8,9], and localized pancreatic atrophy of the pancreas was found in 33% of the patients in this study. From these observations, we consider that it is necessary to encourage other departments that perform routine imaging examinations to be aware of indirect imaging features of early-stage PDAC.

For preoperative pathological diagnosis of early-stage PDAC, the utility of ERCP-associated PJC has been reported, especially SPACE using an ENPD catheter [26,27]. In this study, a single PJC showed a sensitivity of 38%, whereas SPACE showed a significant improvement, with the sensitivity reaching 75%, rising to 83% when considering only patients in stage 0. These results were better than those reported in an analysis of 200 patients with early-stage PDAC reported by the Japanese Study Group on Early Detection of Pancreatic Cancer [20]. One of the reasons is that we perform the examinations according to the common strategies described in the Methods. Regarding complications, PEP occurred in 7% patients, which is similar to the incidence reported previously (3.8–15.1%) [28,29,30]. We have already reported that a 4-Fr ENPD catheter can be used to reduce the risk of PEP [31]. Collection of PJC can be performed in various ways, and we consider that it is a need for complementary use of these methods, depending on the circumstances of pancreatography and abnormal MPD findings.

The sensitivity of EUS-FNA in stage IA was 93%, which was comparable to the sensitivity of EUS-FNA for PDAC including advanced patients (89–92%) [32,33]. We consider that EUS-FNA should be useful for small-diameter PDAC. Meanwhile, recently, the presence of hypoechoic areas around an MPD stenosis has been reported as a characteristic feature of EUS findings in PCIS [34]. Izumi et al. reported that a hypoechoic area surrounding the MPD stenosis was detected in 56.3% of PCIS, and fatty infiltration was detected in 43.8% of PCIS. They also reported that histopathological examination of the periphery of the lesion revealed inflammation and fibrosis in all PCIS [35]. It is conceivable that the hypoechoic lesion could reflect the fibrosis (Figure 2). From these observations, we strongly recommend using not only EUS-FNA, but also ERCP-associated PJC, when EUS shows a hypoechoic lesion surrounding MPD stenosis.

Recently, concerns were raised regarding needle tract seeding by EUS-FNA. Yane et al. reported that out of 176 patients with pancreatic body and tail cancer undergoing preoperative EUS-FNA, 3.4% were diagnosed as having needle tract seeding [36]. A summary report of 15 patients showed that all needle tract seeding after EUS-FNA for diagnosis of PDAC occurred in the trans-gastric route [37]. Therefore, needle tract seeding should be avoided in patients with small PDAC in pancreatic body and tail which are expected to have a long-term prognosis. The revised CGL issued by JPS in 2019 has proposed preoperative chemotherapy for patients with surgically resectable tumors [38], and safe and accurate confirmation of malignancy should be necessary in patients with early-stage PDAC. EUS-FNA may be difficult to perform in patients of suspected PCIS without a tumor, and ERCP-associated PJC may be useful. If chemotherapy before distal pancreatectomy in small PDAC is needed, ERCP-associated PJC should be considered to avoid needle tract seeding. Although the CGL issued by JPS in 2019 has described a diagnostic algorithm for patients, including those with advanced PDAC [13], the results of the present study suggest that a more subtle algorithm based on the size and localization of the tumor and the imaging findings should be used. We would like to propose an effective algorithm for “early diagnosis of PDAC stage 0 and IA with promising long-term prognosis” based on our results. In addition to US and serum pancreatic enzyme testing at medical health check-ups, US should first be performed in patients with risk factors to identify asymptomatic patients efficiently. If an obvious tumor is experienced, meticulous examination according to conventional algorithms should be performed. Patients without a pancreatic tumor and with indirect findings, such as abnormalities of the MPD, cystic lesions, and pancreatic atrophy, should be monitored closely for the early diagnosis of PDAC. In this case, the next step would be to perform MRI including MRCP. If MRI shows any abnormalities of the MPD, EUS should be performed. Regarding pathological examinations in patients having abnormalities without a tumor, ERCP-associated PJC should be performed preferentially for diagnosis of PCIS. If a tumor is detected, ERCP-associated PJC is preferred for the patients with a tumor located in the pancreatic body or tail. ERCP-associated PJC or EUS-FNA is preferred for patients with a tumor located in the pancreatic head (Figure 3).

This study has several limitations. First, a retrospective study design was used. Although this study was a multicenter report, the number of patients was small. Second, regarding EUS and ERCP, no standardized training course is available in Japan. Therefore, these methods are not necessarily used, and effective training courses for young endoscopists should be implemented in the future. Third, although there have been no severe patients, there is still a concern about the problem of PEP.

Meanwhile, the ongoing development of biomarkers using microRNA of pancreatic juice [39] and studies of duodenal juice [40] is expected to improve the accuracy of diagnosis. Complimentary use of imaging findings and newly developed biomarkers will be needed to identify patients with early-stage PDAC.

## 5. Conclusions

Early diagnosis is essential for improving the prognosis of PDAC. The algorithms in the current CGL issued by JPS in 2019 may have some limitations regarding the early diagnosis of PDAC, and new diagnostic algorithms for early diagnosis of PDAC should be developed.

In the future, increased awareness on imaging findings associated with early-stage PDAC and the value of EUS and ERCP in early diagnosis combined with the development of a new diagnostic algorithm for small PDAC should lead to an increased number of diagnosis of early-stage PDAC patients and finally contribute to improving prognosis in patients with PDAC.

## Figures and Tables

**Figure 1 diagnostics-11-00287-f001:**
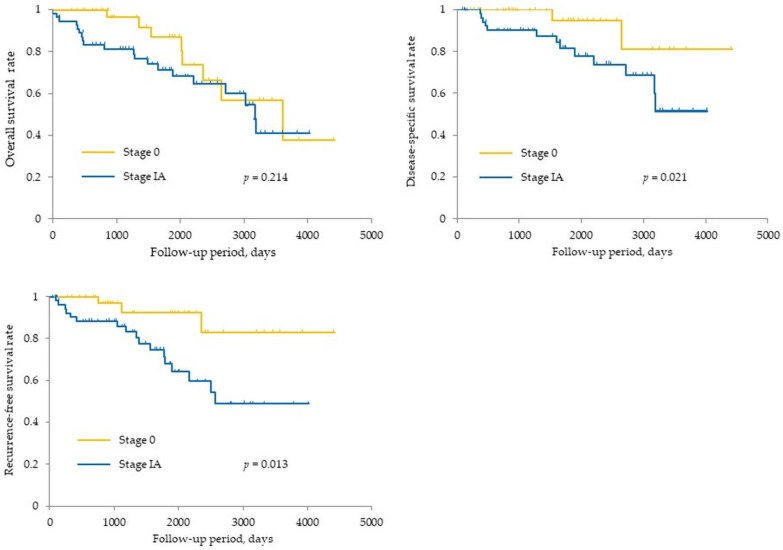
Kaplan–Meier method estimates of overall survival, disease-specific survival, and recurrence-free survival according to the stages.

**Figure 2 diagnostics-11-00287-f002:**
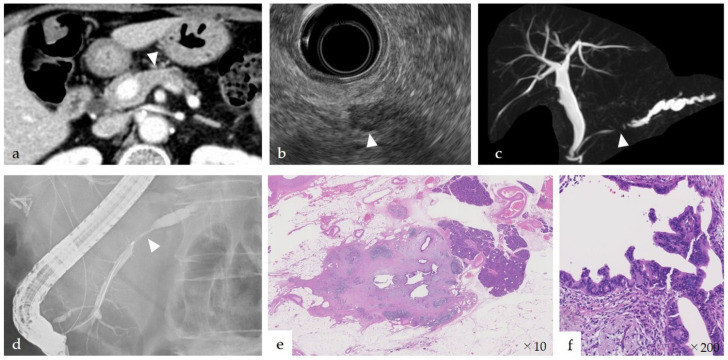
An 82-year-old woman with pancreatic carcinoma in situ (PCIS). (**a**) Localized tissue atrophy was observed in the pancreatic body on computed tomography (arrowhead). (**b**) Endoscopic ultrasonography (EUS) revealed a hypoechoic area, indicating a mass (arrowhead). (**c**) Magnetic resonance cholangiopancreatography revealed that the main pancreatic duct (MPD) was stenotic (arrowhead) in the pancreatic body and the dilated in the caudal part. (**d**) Endoscopic retrograde cholangiography revealed that the MPD was stenotic in the pancreatic body (arrowhead) and dilated in the caudal part. (**e**,**f**) Histopathological findings revealed PCIS in the stenosis of MPD, inflammation and fibrosis surrounding the PCIS, and changes in the fatty tissue in localized regions of the pancreatic parenchyma. This fibrosis corresponded to the hypoechoic area observed using EUS.

**Figure 3 diagnostics-11-00287-f003:**
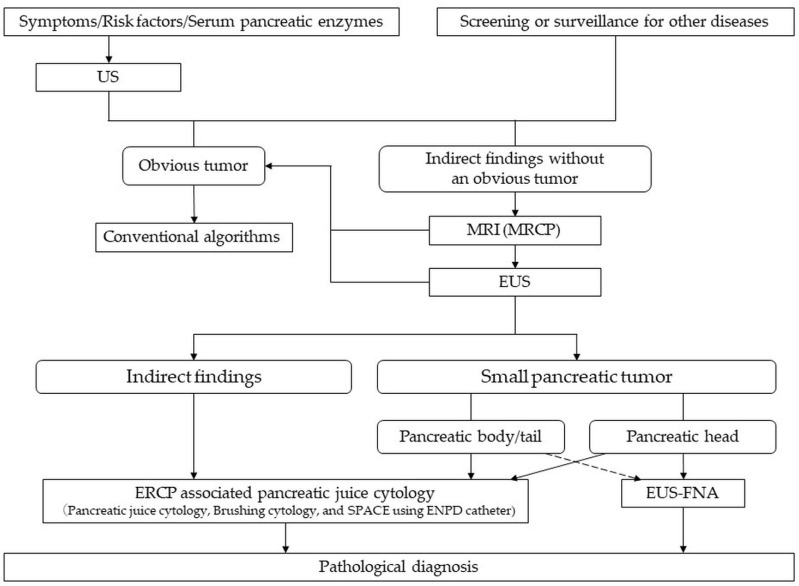
Algorithm for early diagnosis of PDAC at stage 0 and IA with promising long-term prognosis. US, ultrasonography; MRI, magnetic resonance imaging; MRCP, magnetic resonance cholangiopancreatography; EUS, endoscopic ultrasonography; CT, computed tomography; MPD, main pancreatic duct; ERCP, endoscopic retrograde cholangiopancreatography; SPACE, serial pancreatic juice aspiration cytologic examination; ENPD, endoscopic nasopancreatic drainage; EUS-FNA, EUS-guided fine-needle aspiration.

**Table 1 diagnostics-11-00287-t001:** Clinical characteristics of 96 patients with stage 0 and IA pancreatic ductal adenocarcinoma (PDAC).

	All Patients (*n* = 96)	Stage 0 (*n* = 40)	Stage IA (*n* = 56)
Sex (men/women)	47/49	25/15	22/34
Age, mean ± SD (range)	71 ± 9.4 (39–88)	72 ± 8.3 (52–86)	71 ± 10.2 (39–88)
Location, head/body/tail/multiple	30/50/12/4	10/23/6/1	20/27/6/3
Risk Factors, *n* (%) *			
DM	26/96 (27)	8/40 (20)	18/56 (32)
Tobacco use	29/95 (31)	15/40 (38)	14/55 (25)
Heavy alcohol consumption	24/94 (26)	15/39 (38)	9/55 (16)
Obesity (> BMI 30 kg/m^2^)	3/96 (3.1)	2/40 (5.0)	1/56 (1.8)
Past history of acute pancreatitis	6/93 (6.5)	1/39 (2.6)	5/54 (9.3)
Chronic pancreatitis	7/94 (7.4)	5/40 (13)	2/54 (3.7)
IPMN/Pancreatic cyst	27/95 (28)	15/40 (38)	12/55 (22)
Family history of pancreatic cancer	2/84 (2.4)	1/38 (2.6)	1/46 (2.2)
Familial pancreatic cancer	0/96 (0)	0/38 (0)	0/46 (0)
Hereditary pancreatic cancer syndrome	0/96 (0)	0/38 (0)	0/46 (0)
Hepatitis B virus infection	3/93 (3.2)	0/39 (0)	3/54 (5.6)

Data are expressed as number (percentage) or mean ± standard deviation. * Some patients had multiple risk factors. SD, standard deviation; DM, diabetes mellitus; BMI, body mass index; IPMN, intra ductal papillary mucinous neoplasm.

**Table 2 diagnostics-11-00287-t002:** Reasons for medical examination in patients with stage 0 and IA PDAC.

	All Patients (*n* = 96)	Stage 0 (*n* = 40)	Stage IA (*n* = 56)
Symptoms	27/96 (28)	13/40 (33)	14/56 (25)
Abdominal pain/Back pain/Loss weight/Jaundice/Others	18/4/3/1/4	10/1/2/1/2	8/3/1/0/2
Abnormalities identified on medical health check-ups	26/96 (27)	8/40 (20)	18/56 (32)
Abnormal findings on US	18/26 (69)	5/8 (63)	13/18 (72)
Detection of tumors/Indirect imaging findings	4/14	0/5	4/9
Abnormal findings on CT	2/26 (7.7)	0/8 (0)	2/18 (11)
Detection of tumors/Indirect imaging findings	1/1	0/0	1/1
Elevated serum pancreatic enzyme levels	3/26 (11)	2/8 (25)	1/18 (5.6)
Onset or exacerbation of DM	3/26 (11)	1/8 (13)	2/18 (11)
Abnormalities incidentally detected during screening or surveillance for other disease	35/96 (36)	16/40 (40)	19/56 (34)
Abnormal findings on US	4/35 (11)	3/16 (19)	1/19 (5.3)
Detection of tumors/Indirect imaging findings	0/4	0/3	0/1
Abnormal findings on CT	19/35 (54)	9/16 (56)	10/19 (53)
Detection of tumors/Indirect imaging findings	4/15	1/8	3/7
Elevated serum pancreatic enzyme levels	2/35 (5.7)	1/16 (6.3)	1/19 (5.3)
Elevated serum tumor marker level	1/35 (2.9)	0/16 (0)	1/19 (5.3)
Onset or exacerbation of DM	5/35 (14)	1/16 (6.3)	4/19 (21)
Others	4/35 (11)	2/16 (13)	2/19 (11)
Screening primary disease DM/Liver disease/Other cancer/Lung disease/Other	8/6/12/3/6	3/2/5/1/5	5/4/7/2/1
Abnormalities during follow-up of pancreatic disease	6/96 (6.3)	3/40 (7.5)	3/56 (5.4)
Pancreatic cysts/Acute pancreatitis/Chronic pancreatitis	3/1/2	1/0/2	2/1/0
Unknown	2/96 (2.1)	0/40 (0)	2/56 (3.6)

Data are expressed as number (percentage). US, ultrasonography; DM, diabetes mellitus; CT, computed tomography.

**Table 3 diagnostics-11-00287-t003:** Blood tests in patients with stage 0 and IA PDAC.

	All Patients (*n* = 96)	Stage 0 (*n* = 40)	Stage IA (*n* = 56)
Abnormalities of pancreatic enzymes	46/94 (49)	19/39 (49)	27/55(49)
Increased amylase	19/77 (25)	9/31 (29)	10/46(22)
Depressed amylase	5/77 (6.5)	3/31 (9.7)	2/46(4.3)
Increased pancreatic amylase	22/74 (30)	11/37 (30)	11/37 (30)
Depressed pancreatic amylase	3/74 (4.1)	1/37 (2.7)	2/37 (5.4)
Increased lipase level	21/44 (48)	6/13 (46)	15/31 (48)
Depressed lipase	1/44 (2.3)	0/13 (0)	1/31 (3.2)
Increased elastase 1	7/24 (29)	2/6 (33)	5/18 (28)
Abnormal sugar tolerance	25/93 (27)	8/39 (21)	17/54 (31)
Increased tumor markers level			
CEA	6/87 (6.9)	1/37 (2.7)	5/50 (10)
CA19-9	25/94 (27)	4/38 (11)	21/56 (38)
DUPAN-2	9/52 (17)	1/21 (4.8)	8/31 (26)
SPAN-1	6/32 (19)	1/10 (10)	5/22 (23)

Data are expressed as number (percentage). Amylase, amylase secreted from salivary gland and pancreas; Pancreatic amylase, amylase secreted from pancreas; CEA, carcinoembryonic antigen; CA19-9, carbohydrate antigen 19-9; DUPAN-2, duke pancreatic monoclonal anti-gen type2; SPAN-1, s-pancreas-1 antigen.

**Table 4 diagnostics-11-00287-t004:** Imaging findings and modalities in patients with stage 0 and IA PDAC.

	All Patients (*n* = 96)	Stage 0 (*n* = 40)	Stage IA (*n* = 56)
Performed, US/CT/MRI/EUS/ERCP	51/92/75/95/86	13/39/36/40/38	38/53/39/55/48
Detection of pancreatic tumors			
US	20/51 (39)	0/13 (0)	20/38 (53)
CT	37/92 (40)	6/39 (15)	31/53 (58)
MRI	18/75 (24)	3/36 (8.3)	15/39 (38)
EUS	53/95 (56)	8/40 (20)	45/55 (82)
Indirect imaging findings			
MPD dilatation			
US	31/51 (61)	8/13 (62)	23/38 (61)
CT	67/92 (73)	28/39 (72)	39/53 (74)
MRI (MRCP)	55/75 (73)	28/36 (78)	27/39 (69)
EUS	71/92 (77)	31/40 (78)	40/52 (77)
ERCP	56/86 (65)	23/38 (61)	33/48 (69)
MPD stenosis			
MRI (MRCP)	61/75 (81)	31/36 (86)	30/39 (77)
EUS	53/92 (60)	29/40 (73)	24/52 (46)
ERCP	75/86 (87)	31/38 (82)	44/48 (92)
Pancreatic cysts			
US	13/50 (26)	5/13 (38)	8/38 (21)
CT	35/92 (38)	16/39 (41)	19/53 (36)
MRI (MRCP)	37/75 (49)	18/36 (50)	19/39 (49)
EUS	34/92 (37)	16/40 (40)	18/52 (35)
Localized pancreatic tissue atrophy			
CT	30/92 (33)	12/39 (31)	18/53 (34)
EUS	7/84 (8.3)	4/40 (10)	3/52 (5.8)
Hypoechoic area surrounding the MPD stenosis			
EUS	35/92 (38)	18/40 (45)	17/52 (33)

Data are expressed as number (percentage). US, ultrasonography; CT, computed tomography; MRI, magnetic resonance imaging; EUS, endoscopic ultrasonography; ERCP, endoscopic retrograde cholangiopancreatography; MPD, main pancreatic duct.

**Table 5 diagnostics-11-00287-t005:** Preoperative pathological diagnosis in patients with stage 0 and IA PDAC.

	All Patients (*n* = 96)	Stage 0 (*n* = 40)	Stage IA (*n* = 56)
ERCP	86/96 (90)	38/40 (95)	48/56 (86)
Confirmation of malignancy	72/86 (84)	33/38 (87)	39/48 (81)
Single aspiration of pancreatic juice	26/68 (38)	15/34 (44)	11/34 (32)
Brushing cytology of MPD stenosis	14/20 (70)	3/3 (100)	11/17 (65)
SPACE	58/75 (75)	29/35 (83)	29/40 (73)
EUS-FNA	18/96 (19)	3/40 (7.5)	15/56 (27)
Confirmation of malignancy	15/18 (83)	1/3 (33)	14/15 (93)
Cytology	14/18 (78)	1/3 (33)	13/15 (87)
Biopsy	10/18 (56)	1/3 (33)	9/15 (60)
Preoperative confirmation of malignancy	79/96 (82)	33/40 (83)	46/56 (82)

Data are expressed as number (percentage). ERCP, endoscopic retrograde cholangiopancreatography; MPD, main pancreatic duct; SPACE, serial pancreatic juice cytologic examination; EUS-FNA, endoscopic ultrasonography-guided fine-needle aspiration.

## Data Availability

The data presented in this study are available on request from the corresponding author. The data are not publicly available due to privacy.

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
