# Peer review of "Clinical Analysis of Early-Stage Pancreatic Cancer and Proposal for a New Diagnostic Algorithm: A Multicenter Observational Study"

_diagnostics, 2021, doi:10.3390/diagnostics11020287_

Round 1

Reviewer 1 Report

This retrospective, multicenter, long-term observational study aims to suggest a diagnostic protocol based on the analysis of patients who had early (stage 0 and IA) pancreatic ductal adenocarcinoma (PDAC). PDAC is one of the GI caners with the highest mortality rate, and it’s poor prognosis is the consequence of the fact that PDAC is often diagnosed in an progressive, late stage, when the one and only curative surgical intervention can not be done. Finding a diagnostic algorithm which makes possible to diagnose PDAC in an early stage is one of the most important task recently. Therefore, the topic is important, and actual.  

MINOR COMMENTS

1. Introduction: the citation for 5-year survival of PDAC is rather old. The prognosis of PDAC is recently slightly better than 5%. Current reference is needed.

2. Methods:

- I assume that the inclusion of the patients is based on post-operative pathological classification, isn’t it? If yes, it should be clearly stated in the text.

- IPMN was an exclusion criterion of the study. However, PDAC patients with IPMN were enrolled in table 1. How can you solve this contradiction?

- Linear echoendoscope seems to be more appropriate to examine especially the tail of the pancreas. Radial echoendoscope was used in first line during the diagnostical process, and only EUS-FNA was performed with linear echoendoscope. What is the reason of this?

3. Table 2:

- 26 asymptomatic patients were included in the study, in whom abnormalities were identified on medical health check-ups.  Abnormal findings on CT and elevated serum pancreatic enzyme levels were not mentioned in this group of patients in Table 2. What was detected by CT in two of these patients?

- What abnormalities were detected in the 7 out of 35 asymptomatic patients not mentioning in Table 2 during screening or surveillance for other disease?

- What were the abnormal findings on US and CT other than detection of tumors and indirect imaging findings?

- Onset or exacerbation of DM was the main symptom in altogether 10 patients. Could you discuss it? How did you define the exacerbation of DM and did you perform CT or US in these cases? These data seem to be important to recognize the signs of early (stage 0 and IA) PDAC.

- 3rd line from the bottom: 'Follow-up pancreatic disease' can be omitted, since it is mentioned above and detailed in the following line.

- It is stated that 69 PDAC patients were asymptomatic. However, 26 patients  (with abnormalities detected during the medical health check-ups) and 35 patients (with abnormalities incidentally detected during examination or follow-up) and 6 patients (with abnormalities during follow-up of pancreatic disease) does not equal 69 patients. What about the 2 unknown patients? Were the asymptomatic too? Some more detail about these patients are needed or they should be deleted from this table.

4. Some asymptomatic patients had regular follow-up for pancreatic diseases – could you please detail these diseases and the protocol of surveillance?

5. Table 3:

- What is the difference between increased amylase and increased pancreatic amylase or depressed amylase and depressed pancreatic amylase?

- It is a surprise that serum CA 19-9 were elevated in many cases in stage IA tumors, since this tumor marker usually elevates when the tumor is larger than 3 cm. Could you discuss it? How marked was the CA 19-9 elevation in these cases?

6. Table 4:

- Were the cysts part of the tumor (cystic tumors) or just an accidental finding located away from the tumors?

- First line: in stage 0 patients 38 had ERCP and in the IA group 49 (sum: 87), but in the first column „all” only 86 ERCPs are presented. This should be cleared.

- EUS-FNA/FNB is recommended to use instead of EUS-FNA

7. Table 5: What is the cause that the sensitivity of ERCP-associated PJC, especially brushing is better in stage 0 group as compared to that in stage IA group?

8. 'If distal pancreatectomy preoperative chemotherapy in small PDAC is needed,'… Typo may be present in this sentence.

Author Response

We wish to express our appreciation to the Reviewer for the insightful comments, which have helped us significantly improve the paper.

Comment 1. Introduction: the citation for 5-year survival of PDAC is rather old. The prognosis of PDAC is recently slightly better than 5%. Current reference is needed.

 Response 1: In accordance with the Reviewer’s comment, we have changed to the following text. (p. 2, lines 60-61)

“The 5-year survival rate of patients with PDAC was 6% in the USA [2].”

to

“The 5-year survival rate for patients with PDAC is only 6 to 8% in the USA [2].”

We have also changed the following reference.

  1. Rawla, P.; Sunkara, T.; Gaduputi, V. Epidemiology of Pancreatic Cancer: Global Trends, Etiology and Risk Factors. World J. Oncol. 2019, 10, 10-27.

Comment 2. Methods: I assume that the inclusion of the patients is based on post-operative pathological classification, isn’t it? If yes, it should be clearly stated in the text.

Response: In accordance with the Reviewer’s comment. we have changed the following text. (p. 2, lines 83-88)

“ Early-stage PDAC was defined as patients with stage 0 (high-grade pancreatic intraepithelial neoplasia/pancreatic carcinoma in situ [PCIS]) and stage IA (invasive carcinoma with tumor diameter of < 20 mm localized within the pancreas, along with the absence of regional lymph nodes metastasis and distant metastasis) according to the seventh edition of the Japanese Classification of Pancreatic Carcinoma [10].”

to

“Early-stage PDAC was defined as patients with stage 0 (high-grade pancreatic intraepithelial neoplasia/pancreatic carcinoma in situ [PCIS]) and stage IA (invasive carcinoma with tumor diameter of < 20 mm localized within the pancreas, along with the absence of regional lymph nodes metastasis and distant metastasis) based on post-operative pathological classification according to the seventh edition of the Japanese Classification of Pancreatic Carcinoma [10].

Comment 3. Methods: IPMN was an exclusion criterion of the study. However, PDAC patients with IPMN were enrolled in table 1. How can you solve this contradiction?

Response: We apologize for the confusion. We meant to include IPMN concomitant PDAC and exclude high-grade IPMN or IPMN-derived invasive cancer. We have changed the following text. (p.2, lines 88-91)

“Patients with high-grade intraductal papillary mucinous neoplasm (IPMN) or IPMN-derived invasive cancer showing a histologic transition between IPMN and PDAC were excluded.”

to

“Patients with intraductal papillary mucinous neoplasm (IPMN) concomitant PDAC were included, but patients with high-grade IPMN or IPMN-derived invasive cancer showing a histologic transition between IPMN and PDAC were excluded.”

Comment 4. Methods: Linear echoendoscope seems to be more appropriate to examine especially the tail of the pancreas. Radial echoendoscope was used in first line during the diagnostical process, and only EUS-FNA was performed with linear echoendoscope. What is the reason of this?

Response: We used radial echoendoscope in first line because we thought that radial echoendoscope is superior for observing the pancreatic duct in the long axis. Indeed, linear echoendoscope may be superior for observation of the tail of the pancreas. We used linear echoendoscope in some cases. we have changed the following text. (p.3, lines 107-112)

“For endoscopic ultrasonography (EUS), we used a radial echoendoscope (GF-UE260 and GF-UM2000; Olympus Medical Systems, Tokyo, Japan) equipped with processors (EU-ME1and EU-ME2; Olympus Medical Systems, α-10 and F75; Hitachi-Aloka, Tokyo, Japan).”

to

“For endoscopic ultrasonography (EUS), we used a radial echoendoscope (GF-UE260 and GF-UM2000; Olympus Medical Systems, Tokyo, Japan), which was superior for observing the MPD in the long axis, equipped with processors (EU-ME1and EU-ME2; Olympus Medical Systems, α-10 and F75; Hitachi-Aloka, Tokyo, Japan). If the pancreatic tail was poorly observed, linear echoendoscope (UCT-260; Olympus Medical Systems) was also used as needed.”

Comment 5. Table 2: 26 asymptomatic patients were included in the study, in whom abnormalities were identified on medical health check-ups.  Abnormal findings on CT and elevated serum pancreatic enzyme levels were not mentioned in this group of patients in Table 2. What was detected by CT in two of these patients?

Response: The two patients who had abnormalities identified on CT of medical health check-ups were both stage IA, one with detection of tumor and the other with indirect imaging findings. We have added CT findings and pancreatic enzyme levels of medical health check-ups in Table 2 and added following text. (p. 4, 175-177)

“The abnormal findings detected by CT included a pancreatic tumor in one patient and indirect imaging finding in one patient.”

Comment 6. Table 2: What abnormalities were detected in the 7 out of 35 asymptomatic patients not mentioning in Table 2 during screening or surveillance for other disease?

Response: Two patients had abnormal pancreatic enzyme levels. One patient had elevated tumor markers, and the above three patients were added to Table 2. The abnormal findings detected by CT included tumors in four patient and indirect imaging findings in 15 patients. Of the other four patients, three patients had abnormalities noted on imaging examination other than CT or MRI, and one patient diagnosed PCIS in the postoperative specimen performed for cholangiocarcinoma. We have changed the following text. (p.5, lines 178-186)

“Among the 35 patients in whom any abnormalities were incidentally detected during examination or follow-up for other diseases, 23 had abnormal findings on CT (n = 19) or on US (n = 4). Of the abnormal imaging findings, tumors were detected in four patients, while indirect imaging findings were detected in 19 patients (Table 2).”

to

“Among the 35 patients in whom any abnormalities were incidentally detected during examination or follow-up for other diseases, two (5.7%) were detected by US, 19 (54%) were detected by CT, two (5.7%) exhibited elevated serum pancreatic enzyme levels, and one patients (2.9%) exhibited elevated serum tumor marker level. All of the abnormal findings detected by US were indirect imaging findings. The abnormal findings detected by CT included tumors in four patient and indirect imaging findings in 15 patients. Of the other four patients, three patients had pancreatic abnormalities noted on imaging examination other than US or CT, and one patient was diagnosed with PCIS in the post-operative specimen performed for extrahepatic cholangiocarcinoma.”

Comment 7. Table 2: What were the abnormal findings on US and CT other than detection of tumors and indirect imaging findings?

Response: The abnormal findings on US and CT were either detection of tumor or indirect imaging findings (pancreatic cysts, pancreatic duct dilatation pancreatic duct stenosis, and localized pancreatic atrophy).

Comment 8. Table 2: Onset or exacerbation of DM was the main symptom in altogether 10 patients. Could you discuss it? How did you define the exacerbation of DM and did you perform CT or US in these cases? These data seem to be important to recognize the signs of early (stage 0 and IA) PDAC.

Response: We apologize for incorrect notation. Three patients were defined the onset or exacerbation of DM during medical health check-ups, and onset or exacerbation of DM was the main symptom in altogether eight patients. Exacerbation of DM was defined as an increase in HbA1c of 1% or more within six months. CT and US were performed on these patients.

We have added the following text (p.9, line 286):

“Exacerbation of DM was defined as an increase in HbA1c of 1% or more within six months.”

Comment 9. Table 2: 3rd line from the bottom: 'Follow-up pancreatic disease' can be omitted, since it is mentioned above and detailed in the following line.

Response: In accordance with the Reviewer’s comment, we have omitted the line.

Comment 10. Table 2: It is stated that 69 PDAC patients were asymptomatic. However, 26 patients (with abnormalities detected during the medical health check-ups) and 35 patients (with abnormalities incidentally detected during examination or follow-up) and 6 patients (with abnormalities during follow-up of pancreatic disease) does not equal 69 patients. What about the 2 unknown patients? Were the asymptomatic too? Some more detail about these patients are needed or they should be deleted from this table.

Response: We thank the reviewer for this pertinent suggestion. Two patients were unknown for their reason for medical examination. We have re-examined in 67 asymptomatic patients.

Comment 11. Some asymptomatic patients had regular follow-up for pancreatic diseases – could you please detail these diseases and the protocol of surveillance?

Response: In accordance with the Reviewer’s comment, we have added the following text. (p. 5, lines 187-191)

“Six patients were detected pancreatic abnormalities during follow-up for pancreatic disease, including pancreatic cysts, acute and chronic pancreatitis. All patients had been followed up every six months with either CT, EUS, or MRI and measurement of serum tumor markers including carcinoembryonic antigen (CEA) and carbohydrate antigen 19-9 (CA19-9) (Table 2).”

Comment 12. Table 3: What is the difference between increased amylase and increased pancreatic amylase or depressed amylase and depressed pancreatic amylase?

Response: Amylase is the total amount of amylase secreted from salivary gland and pancreas. Pancreatic amylase is secreted from pancreas. Since some institutions measured only amylase, some measured only pancreatic amylase, and some measured both, amylase and pancreatic amylase were listed separately.

Comment 13. Table 3: It is a surprise that serum CA 19-9 were elevated in many cases in stage IA tumors, since this tumor marker usually elevates when the tumor is larger than 3 cm. Could you discuss it? How marked was the CA 19-9 elevation in these cases?

Response: We appreciate the Reviewer’s comment. It has been reported that there is no correlation between the tumor diameter and serum CA19-9 level except for PDAC with a diameter of more than 6 cm. Then, we have added the following text (p.10, lines 289-293):

“However, serum CA19-9 were elevated in 38% patients in stage IA in this study. It has been reported that there was no correlation between the tumor diameter and the serum CA19-9 level except for PDAC with a diameter of more than 6 cm [18], and it is considered necessary to measure serum CA19-9 level even for a small pancreatic tumor.”

 We have also added the following reference.

18.Jung, K. W.; Kim, M. H.; Lee, T. Y.; Kwon, S.; Oh, H. C.; Lee, S. S.; Seo, D. W.; Lee, S. K. Clinicopathological aspects of 542 cases of pancreatic cancer: a special emphasis on small pancreatic cancer. J. Korean Med. Sci. 2007, 22, S79-85.

Comment 14. Table 4: Were the cysts part of the tumor (cystic tumors) or just an accidental finding located away from the tumors?

Response: Cysts were just an accidental finding located away from the tumors.

Comment 15. Table 4: First line: in stage 0 patients 38 had ERCP and in the IA group 49 (sum: 87), but in the first column „all” only 86 ERCPs are presented. This should be cleared.

Response: As you pointed out, it is a mistake in the description. It was 86 patients (38 patients with stage 0 and 48 patients with stage IA) in total. We have revised the line in table 4.

Comment 16. Table 4: EUS-FNA/FNB is recommended to use instead of EUS-FNA

Response: We appreciated the Reviewer’s comment. EUS-FNB is ideal, but there were many cases that FNB could not be performed due to small tumor size, so we would like to use “EUS-FNA”.

Comment 17. Table 5: What is the cause that the sensitivity of ERCP-associated PJC, especially brushing is better in stage 0 group as compared to that in stage IA group?

Response: Indeed, the sensitivity of ERCP-associated PJC, especially brushing cytology, of the stage 0 was better than stage IA. However, the number of cases was very small and there was no significant difference in sensitivity between stage 0 and IA. In the future, we will accumulate cases and investigate the cause.

Comment 18. 'If distal pancreatectomy preoperative chemotherapy in small PDAC is needed,'… Typo may be present in this sentence.

Response: We appreciate the Reviewer’s comment, we have changed the following text. (p. 11, 372-374)

“If distal pancreatectomy preoperative chemotherapy in small PDAC is needed, ERCP-associated PJC could be considered to avoid needle tract seeding.”

to

“If chemotherapy before distal pancreatectomy in small PDAC is needed, ERCP-associated PJC should be considered to avoid needle tract seeding.”

Thank you again for your comments on our paper. We trust that the revised manuscript is suitable for publication.

Reviewer 2 Report

Multicenter study with impressive number (96) of patients with pancreatic adenocarcinoma (PDAC) in stage 0/ 1A, to clarify the clinicopathological features, and to propose new algorithm for early diagnosis of PDAC, mostly asymptomatic. Well designed, large amount of data and new data. Including procedures not widely available.

Shoud be published. However Discussion is too long (line 250-444) and from line 250 to 406 should definitely be reduced (at least by 30%) but leaving 407-444.

Minor change: line 404-5 please, make it clear.

Author Response

We wish to express our appreciation to the Reviewer for the insightful comments, which have helped us significantly improve the paper.

1. Discussion is too long (line 250-444) and from line 250 to 406 should definitely be reduced (at least by 30%) but leaving 407-444.

 Response 1: We thank the Reviewer for this pertinent comment. In accordance with the Reviewer’s comment, we have reduced Discussion.

Minor change: line 404-5 please, make it clear.

 Response 2: We appreciate the Reviewer’s comment. We have omitted that text due to reduce Discussion.

Thank you again for your comments on our paper. We trust that the revised manuscript is suitable for publication.